# Use of Multi-Agent Theory to Resolve Complex Indoor Air Quality Control Problems

**DOI:** 10.3390/s19051206

**Published:** 2019-03-09

**Authors:** Shang-Yuan Chen, Cheng-Yen Chen

**Affiliations:** School of Architecture, Feng Chia University, No. 100, Wenhwa Rd., Seatwen, Taichung 40724, Taiwan; archschool@fcu.edu.tw

**Keywords:** intelligent agent, indoor air quality, fine particulate matter, conflict resolution program

## Abstract

Taiwan has suffered from widespread haze and poor air quality during recent years, and the control of indoor air quality has become an important topic. This study relies on Multi-Agent theory in which collected air quality was used in calculations and after agents make decisions in accordance with pre-written rules to construct and indoor air quality control system and conflict resolution mechanism, which will serve to maintain a healthy and comfortable indoor environment. As for implementation, the simulated system used the Arduino open source microcontroller system to collect air quality data and turn on building equipment in order to improve indoor air quality. This study also used the graphic control program LabVIEW to write a control program and user interface. The implementation verifies the feasibility of applying multi-agent theory to air quality control systems, and an Individual intelligent agent has the basic ability to resolve their own conflicts autonomously. However, when there are multiple factors and user status are simultaneously involved in the decision-making, it is difficult for the system to exhaust all conflict conditions, and when context control surpassing the restrictions of binary logic rule-based reasoning, it is necessary to change the algorithm and redesign the system.

## 1. Motivation and Goal

Taiwan’s economy has grown rapidly over the past few decades, but although this has greatly increased wealth, it has also caused environmental pollution. Air quality control has become an important research issue in an intelligent environment field [1,2]. In particular, because of the topography and prevailing wind directions in western Taiwan, particulate matter produced by industrial parks and thermal power plants cannot easily disperse, which has caused the air pollution situation to deteriorate steadily. Figure 1 shows NASA satellite aerosol optical depth (AOD) data, where aerosol optical depth is a key physical measure of the turbidity of the atmosphere. In this image, the deeper the color, the greater the difficulty light has in penetrating. It can be seen that there are large amounts of particulate matter in western and southern Taiwan, and the Yunlin Sixth Naphtha Cracker and Taichung Thermal Power Plant may be major sources of this pollution [3].

While it was widely believed in the past that intelligent buildings were energy-intensive and required large amounts of active equipment, in intelligent building that truly pursues sustainability must employ on “actively thinking” equipment to remedy the inefficiencies of passive buildings, achieve effective energy conservation, and maintain a comfortable living environment. From another perspective, because it is hard for people to pay attention to things continuously for long periods and people are not as sensitive as sensors which can aware the values harmful to the human body, therefore assistance from artificial intelligence is needed for the routine active control of indoor air quality. Intelligent agents and multi-agent societies are control systems relying on artificial intelligence to perform periodic sensing of the environment and take necessary corrective actions. This study constructed a multi-agent indoor air quality control system, and also developed a modeling system to analyze the indoor air quality control system’s feasibility.

## 2. Literature Review

In the ubiquitous computing era of the artificial intelligence (AI), the Internet of Things (IoT), and the booming development of big data technologies, how to build an agent-based smart living space which is capable of supporting sustainable energy conservation as well as maintaining health and comfortable environment for housing occupants has become an important research issue [4,5,6]. Autonomous intelligent agents and their society is capable of sensing, calculating, actuating, and communicating. The major dimensions of research include: (1) Perception: Various environmental sensors, and signal processing [7,8,9]. (2) Computation: Algorithms include binary logic rule-based reasoning, expert system, artificial neural network learning algorithm, fuzzy neural network algorithm capable of reasoning and learning, and other optimization algorithms [10,11]. (3) Actuation: Agent behavior patterns include: An individual agent’s autonomous behaviors, including re-action, pro-action, and inter-active. As well as the relationships of the agent society, including hierarchy, homogeneity, heterogeneity and complementarity [12]. And actuators, including various building components and environmental control equipment [13]. (4) Communication: Transmission methods include wired network, wireless network [8,9], micro-grids [14], smart grids [15], and communication protocol. (5) Integrated applications include: Programmable Controller, user interface (UI) design [16,17], intelligent building design [18,19], planning for smart city. And other derivative issues include: E-commerce applications [20], information security [20], optimization management [21], post-use evaluation [22], verification of system validity and reliability [23], and more.

This paper advocates building multi-agent based smart living space with pro-active building components and environmental control equipment to improve the insufficient performance of a passive building and to achieve the maintenances of the purposes of sustainable energy conservation and occupants’ health and comfort simultaneously. Intelligent agents and their society theory and indoor air quality indicators and guidelines are described as follows:

### 2.1. Intelligent Agent Theory

In the field of artificial intelligence, intelligent agents refer to autonomous entities that can sense the surrounding environment and initiate actions to achieve design goals. An agent can receive various types of information from the environment, confirm the order of task priority and target behavior, and make decisions without any need for human intervention. Intelligent agent technology is currently one of the most important research paradigms in the field of artificial intelligence, and focuses on the development and use of knowledge representation tools as computing and information communication mechanisms in an environment characterized by distributed intelligence [12]. The basic modules forming the core of an intelligent agent consist of sensors, computing mechanisms, and actuators, and the agents include both software and hardware. Intelligent agents can assume the role of humans in a system, and help humans judge the state of an environment. A basic and simple reflex agent senses the environment; It can perform actions in accordance with behavioral rules [24] (Figure 2). Furthermore, program developers can use graphic symbols to draft the operating processes of an intelligent agent system, and thereby clarify the operations at each step during system development. Some of the graphic symbols that may be used are shown in Figure 3. As shown by intelligent agent behavior model in Figure 4, an intelligent agent performs calculations after sensing the environment, and relies on actuators to change the environment based on its calculation results [25].

Multi-agents are object-oriented cooperative distributed problem solving systems (OOCDPSS) based on agent societies and comprise the entirety of the various relationships between agents [26]. This framework establishes an inward/outward communication order in an agent community, where a leader agent (LA) serves as the community’s chief responsible agent. When an OOCDPSS deals with a problem, it will divide the problem appropriately into multiple sub-problems, assign the sub-problems to appropriate problem-solvers for resolution, and finally integrate the results of problem solving in order to achieve an appropriate resolution [27]. This framework can be applied in the design of smart building systems, and designers can assign sensing results from different sensors to individual agents for subsequent handling. A communication platform can be used to resolve conflicts between individual agents, and the system can then activate various kinds of equipment. This type of system thus operates in a cycle of sensing, calculation, communication, and action (Figure 5).

### 2.2. Indoor Air Quality

In accordance with Taiwan’s Environmental Protection Administration (EPA), Executive Yuan, indicators of indoor air quality include ozone (O_3_), fine particulate matter (PM_2.5_), particulate matter (PM_10_), carbon monoxide (CO), sulfur dioxide (SO_2_), and nitrogen dioxide (NO_2_). Because of the differences in units used to quantify these substances, in order to simplify sensor measurements and program control criteria, this study employs fine particulate matter (PM_2.5_) as the chief control variable.

Fine particulate matter (PM_2.5_) causes respiratory tract and cardiovascular disease, including myocardial infarction, cerebrovascular disease, and lung cancer [28]. The “Air Quality Guidelines” issued by the World Health Organization in 2005 give a 24-h average PM_2.5_ value standard of PM_2.5_ < 25 μg/m^3^, and an annual average value standard of PM_2.5_ <10 μg/m^3^ [29]. For its part, Taiwan’s EPA provides the air quality indicator grades for fine particulate matter shown in Table 1. Because PM_2.5_ is harmful to human health when its concentration level reaches grade 6, this study therefore recommends that the lower limit of grade 6—a PM_2.5_ concentrations of 48 μg/m^3^—be taken as a threshold value dividing good and poor air quality, and the upper limit of grade 6—53 μg/m^3^—be taken as a buffer value ensuring that particulate matter does not immediately reach a high hazard value after exceeding the good air quality threshold.

## 3. Theory and Method

In accordance with the foregoing analysis, the “indoor air quality control system” constructed in this study required the two dimensions of: (1) a system framework, (2) rule table design.

### 3.1. System Framework

The conditional (if, then) reasoning framework shown in Figure 6 was compiled based on air quality factors and control methods. Individual intelligent agents will then initiate the actions of the building envelope and equipment intended to improve indoor air quality based on air quality monitoring data and if, then decisions. The system operates repeatedly based on the preset detection frequency f; as soon as the building envelope or equipment is activated, the system will operate for the preset duration Ta.

Taking PM_2.5_ fine particulate matter as the chief air quality control variable, this study constructed the simulated environment shown in Figure 7. In this environment, the main computer receives data transmitted from outdoor and indoor sensors, and activates devices based on pre-written criteria after the computer’s intelligent agents perform situation analysis, in a cycle of sensing, calculation, and action.

Our indoor air quality control system based on intelligent agent theory is composed of three parts, namely the input end (sensing), processing end (calculation), and output and (action) (Figure 8), which are described as follows:(1)Input end (sensing):The input end is the source of the air quality control system framework’s data, which comprises indoor environmental data, and outdoor environmental data. After collecting data, the input end transmits it to the processing end.(2)Processing end (calculation):After the collection of indoor and outdoor data, the processing end performs assessment and comparison. It employs a pre-written rules program to determine whether the environment currently has any problems, and whether any facilities or equipment must be activated to improve the environment.(3)Output end (action):The output end serves to activate facility and equipment in order to improve indoor air quality.Implementation of this indoor air quality control system required electromechanical equipment and control software; our system implementation employed Arduino [31] and LabVIEW [32] technologies. The device linkage of an intelligent agent system possessing sensing/calculation/action capabilities can be as shown in Figure 9, where the sensing end chiefly relies on Arduino to operate electromechanical equipment and communicate with the computer. Apart from this, the system uses a gateway to obtain outdoor particulate matter data provided by the environmental protection agency (EPA) from a cloud database. A LabVIEW program is responsible for performing data calculations. At the actuation end, after the program has processed the data, control signals are sent to actuation equipment and devices via Arduino boards in order to improve indoor air quality. The system will maintain one set of actions for time Ta, and begin the next cycle after the sensors have performed sensing with frequency f (Figure 10).

### 3.2. Rule Table Design

An individual intelligent agent with the foregoing sensing-calculation-action ability must have the basic ability to resolve self-conflicts. For instance, if indoor air quality is poor, the windows must be opened to allow ventilation. However, if outdoor air quality is also poor, the agent must close the windows, and turn on the air purifier. This suggests some basic principles for the design of a sustainable building: Employ active adjustment of building elements as much as possible in order to comply with the operating principles of a passive building, and seal the building interior and activate energy-consuming active equipment in order to maintain indoor air quality only when conflicts occur.

In order to investigate the feasibility of the use of agent societies, this study designed the multi-agent framework shown in Figure 11, and added temperature as a control variable. We found that when there are multiple intelligent agents that may control the same device, conflicts may arise due to inconsistent actions on the part of different agents. For instance, when the outdoor air is good (1), but the indoor air quality is poor (0), the agents will naturally decide to open the windows (on) to obtain ventilation. However, when the indoor temperature is excessively high (2), the situation will call for closing the windows (off) and also turning on air conditioning (on), which means that agents must have the ability to communicate with each other.

Multi-agent societies must have “sensing-calculation-communication-action ability”. As shown in Figure 12, intelligent agent 1 receives data from indoor and outdoor particulate matter sensors (when PM_2.5_ concentration ≦ 48 μg/m^3^: air quality good (1), when PM_2.5_ concentration > 48 μg/m^3^: air quality poor (0)), and can control the window-opening device and air purifier (open windows (on), close windows (off)), while intelligent agent 2 receives data from the indoor temperature sensor (when temperature > 26 °C, excessively high (0), when 26 °C ≥ temperature ≥ 20 °C, comfortable (1), when temperature < 20 °C, excessively low(2)), and can control the window-opening device and air conditioning equipment. The two intelligent agents can jointly generate 12 rules (4 × 3 = 12). 

However, because the two intelligent agents can both control the window-opening device, they may have conflicts (shown by dotted lines in the figure), and must therefore be able to communicate and coordinate with each other. There are six situations where conflicts may arise concerning whether to open the windows, in which the two agents make conflicting decisions concerning whether to open the window, and conflict resolution methods must therefore be written into the if-then scenario rules. Each piece of equipment must have an operating time T_an_, and the sensors must continue to monitor the environment at a frequency f after changes to the environment are made.

## 4. Implementation

The following is an overview of the process of verifying the foregoing indoor air quality control system, including the three aspects of: (1) establishment of a multi-agent program, (2) user interface, (3) program implementation and testing:

### 4.1. Establishment of a Multi-Agent Program

This step involved the use of LabVIEW to write a program controlling sensing, judgment, and actuation mechanisms. In this program, when a sensor receives data, as soon as the Yes or No voltage exceeds a threshold value, classification is performed according to a rule table, and various building equipment will be activated. Because it was found through experimentation that it is difficult to control the ambient particulate concentration at a specific value, to ensure that the program and equipment were able to function normally, this study employed a variable resistance slider to simulate input voltage values from indoor and outdoor sensors. As described above, based on a single-agent framework, we added a second agent, as well as a protocol for resolving conflicts between the two agents, to create a multi-agent framework. Because conflicts might occur between the two agents, it was necessary to establish a channel of communication. The conflict resolution model could consist either of transmission of one agent’s judgment to the other agent, and allowing the second agent to perform a summary judgment, or transmission of data from the two agents to a third agent, and allowing the third agent to make a decision. Communication between the agents was determined by the system’s communications protocol. However, since this experiment included the two agents’ rule tables within the same program, we did not have to consider the question of communication for the time being. If the system contained two or three programs, it would be necessary to consider how the programs transmitted data to each other.

Figure 13 shows a full overview of the LabVIEW multi-agent program, which includes a sensing portion (Figure 14), processing and communication portion (left and center of Figure 15 and Figure 16), and actuating portion (right of Figure 15). The sensing portion of the multi-agent program shown in Figure 14 consists of Agent 1, which receives particulate matter data from indoor and outdoor sensors, and Agent 2, which bears responsibility for receiving data from an indoor temperature sensor.

The upper half of Figure 15 shows Agent 1’s PM_2.5_ judgment and control portion, while the lower half shows Agent 2’s temperature judgment and control portion. The rectangular shape in the center of the diagram shows the priority rule conflict resolution protocol used in this experiment and the far right consists of the building equipment control program portion. After receiving indoor and outdoor PM2.5 data, the system decides whether to open windows and turn on the air purifier. And after the temperature sensor transmits indoor temperature data, the system decides whether to open windows and turn on air conditioning equipment. Lastly, the system’s priority rules are employed to resolve any conflicts involving the window-opening device, and determine the sequence in which building equipment will be activated.

The intelligent agents in this study relied on LabVIEW’s case structure mechanism to determine the result of if-then rules. The system depended on input conditions for the activation of different events to make rule-based judgments, and finally activated building equipment in accordance with the rules, with the event framework shown in Figure 16. In the conflict resolution protocol used in this experiment, preset priority rules determined which agent’s judgment had priority. Consequently, when a conflict occurred, the agent with the highest priority obtained the right to control of the equipment in question. Because the two intelligent agents in this experiment were written in the same program, they used local variables to transmit data and perform calculations, and also relied on preset priorities to ensure that conflicts did not occur. However, if the two intelligent agents had been written as separate programs, it would be necessary to consider the communications protocol by which the different programs exchanged data.

### 4.2. User Interface

The user interface consisted of:(1)A sensor status display:The concentration value obtained by the PM_2.5_ sensor could not be displayed directly on the LabVIEW interface, and it was necessary to use a function to convert the sensor’s concentration value to a voltage value for display. As shown in the lower right portion of Figure 17, the air quality threshold is set at 48 μg/m^3^, which is converted to a voltage of 0.871 V. As a consequence, a voltage greater than 0.871 V indicates that a high concentration of pollutants has been detected, and indoor air quality is poor; in contrast, a voltage lower than 0.871 V indicates that indoor air quality is good [33].(2)Actuator status display:The upper portion of Figure 18 shows the status of the window-opening device, including the current opening angle of the servo motor (0°~180°), and whether the air purifier has been turned on.(3)Rule status:As seen in the upper left part of Figure 18, Agent 1 and Agent 2 have a total of 12 rules (4 × 3 = 12), and six of these rules, which concern window operation, may be subject to conflicts (Figure 19). The current status of rule implementation is displayed in this area. Taking Rule 3, for which the light is on, as an example, this rule indicates that If outdoor air quality is poor (0), but indoor air quality is good (1), and indoor temperature is excessively high (0), Then a control command of 0 is transmitted, which represents closing the window, and a fan control code 0 is transmitted via Arduino, which represents air purifier off. All of these rules are written in LabVIEW’s rule-based case structure.

### 4.3. Program Implementation and Testing

Normal operation of this program verified the feasibility of an intelligent agent framework, and showed that the agents could resolve conflicts and control the buildings equipment in accordance with priority. To link Arduino with the computer, the communications file LIFA_Base.ino from the LabVIEW plug-in LabVIEW for Arduino was uploaded to the Arduino device, which allowed Arduino to communicate with LabVIEW. Arduino was connected with 3 slider-controlled variable resistors, which represented 3 sensor values, one servo motor, which represented the status of the window-opening device, one 12V fan, which represented the status of the air purifier, and one row of RGB LED lamps, which respectively indicated that the indoor temperature was excessively high (R), suitable (G), or excessively low (B), in which case the air conditioner would take appropriate action (cooling, off, heating). As shown in Figure 20, a servo motor status of 90° indicated that the window was open, while a status of 0° indicated that the window was closed, and the fan’s on/off status indicated the air purifier’s on/off status. Because Arduino’s 5 V different from the fan’s voltage, a relay was used to control the fan’s on/off status.

## 5. Conclusions

This study constructed an indoor environment containing a multi-agent community with “sensing-calculation-communication-action” ability, which was used to verify the feasibility of applying multi-agent theory to air quality control systems. The experiment found that when only a few pieces of building equipment and a small number of air quality factors are considered, rules can be written in advance to resolve equipment conflicts. However, when there are many types of building equipment and air quality factors, if users intervene in the system’s decision-making, the system will not deal with all possible statuses. As a consequence, the system must allow program revisions and two-way communication with users, or it may be necessary to consider the establishment of a system algorithm, which is discussed as follows:Giving intelligent agents the ability to autonomously resolve problems:Individual intelligent agents with “sensing-calculation-action” ability also have the basic ability to resolve their own conflicts. When single agents have only one sensing criterion, conflicts will not occur. But if agents have an even number of sensing criteria, consideration must be paid to the use of third devices to resolve conflicts. For instance, in the case of Agent 1 in this study, when window-operation commands based on indoor and outdoor sensor data were in conflict, the system could choose to close the window and use a third piece of equipment, namely an air purifier, to resolve the conflict. And because Agent 2 had only one sensing criterion, conflicts could not occur.Use of an agent community to resolve conflicts:When rules are in conflict, the following methods are commonly used to achieve a resolution:(1)Setting the priority order(2)According to the longest match principle(3)Adding new rules [34]This study employed the method of setting a priority order. Taking this experiment as an example, the situation of the agent lacking a resolution method was considered as a first priority. For instance, when the particulate matter concentration was high, the system could use natural ventilation or use the air purifier to resolve the problem. However, because the only way to resolve temperature anomalies was to close the window and turn on the air conditioner, temperature anomalies lacking a resolution method receive priority consideration. The priority order principles for other settings, including those conditions posing an immediate hazard to humans and therefore receiving priority consideration, such as when strong winds or torrential rainfall occurred outside, the system prioritized closing the windows (off). This resolution model required the use of a rule table listing all possible situations, however, and after finding all possible conflicting conditions, also required a “superior” rule table to resolve conflicts. Such a system will be ineffective when there are too many intelligent agents.When situational control transcending the restrictions of binary logic rule-based reasoning:This study used clear-cut threshold values as sensor settings. However, when sensor values are connected with different users’ impressions, the threshold values will become vague, such as when different users have impressions of cold, suitable, and hot temperatures. Furthermore, the equipment operating criteria in this study were very simple, and consisted solely of off and on status. However, windows can open through a range of from 0° to 90°, and air purifiers and air conditioners can be run and strong, medium, and gentle modes, or have different concentration operating settings, or there may be different temperature setting modes. The foregoing binary conditions must be resolved using other calculation methods, such as through the use of fuzzy logic or a neural network. Otherwise, binary logic rules will not be able to take all possible situations into consideration.

## 6. Patents


(1)Air quality control system, M525429, Republic of China, Utility model patent, 2016/0711~2026/0407.(2)Cloud-based air quality control system, M536335, Republic of China, Utility model patent, 2017/0201~2026/1013.


## Figures and Tables

**Figure 1 sensors-19-01206-f001:**
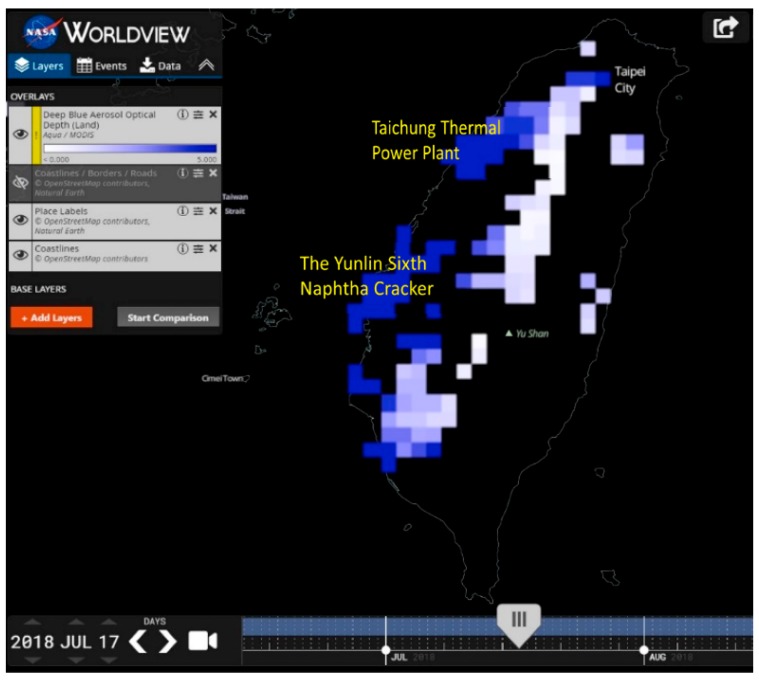
Aerosol optics depth (AOD) satellite image (17 July 2018) [3].

**Figure 2 sensors-19-01206-f002:**
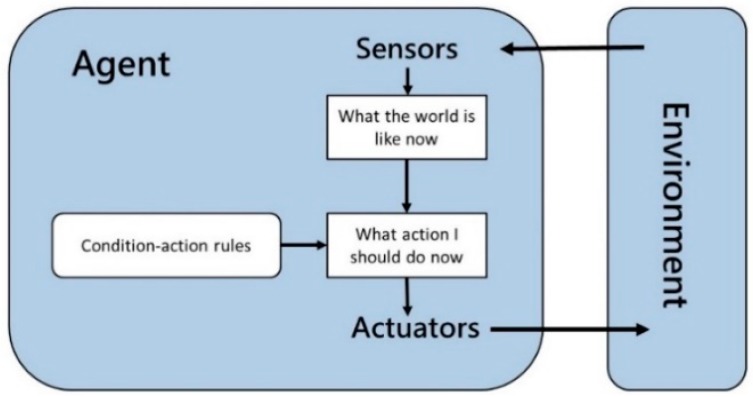
A simple reflex agent-operating model.

**Figure 3 sensors-19-01206-f003:**
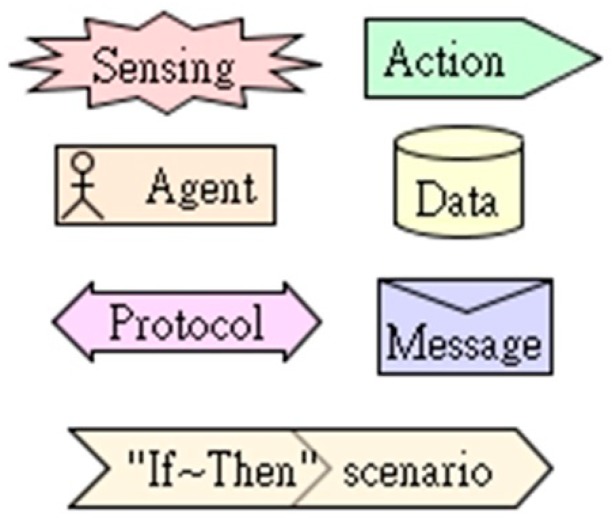
Graphic symbols for intelligent agent systems.

**Figure 4 sensors-19-01206-f004:**
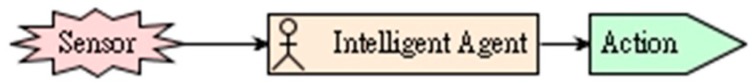
An intelligent agent operating model.

**Figure 5 sensors-19-01206-f005:**
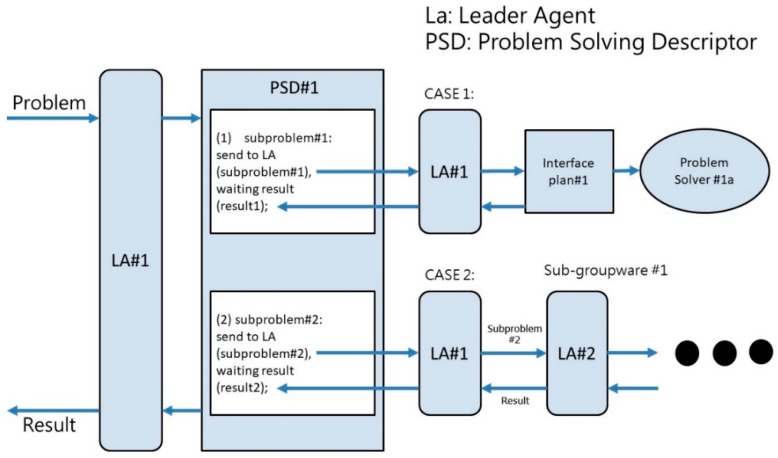
Schematic diagram of OOCDPSS [27].

**Figure 6 sensors-19-01206-f006:**
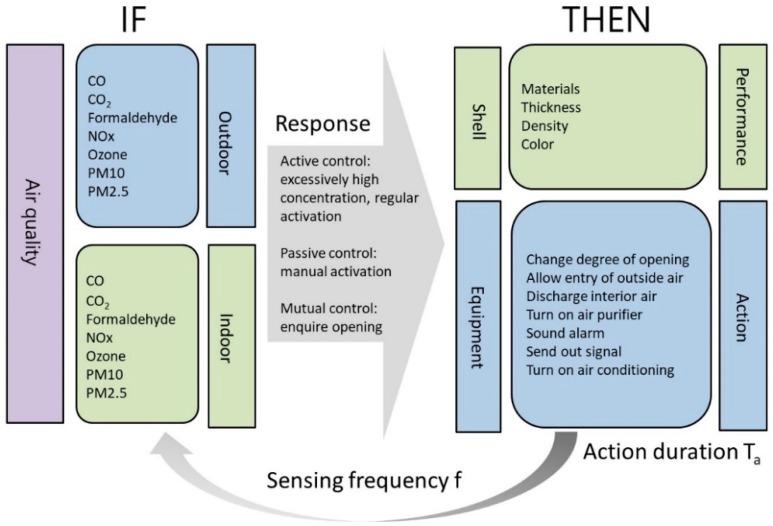
Factors influencing air quality and control methods.

**Figure 7 sensors-19-01206-f007:**
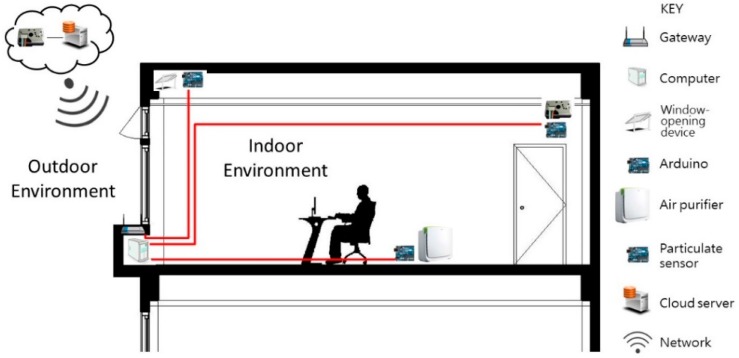
Schematic diagram of the environment.

**Figure 8 sensors-19-01206-f008:**
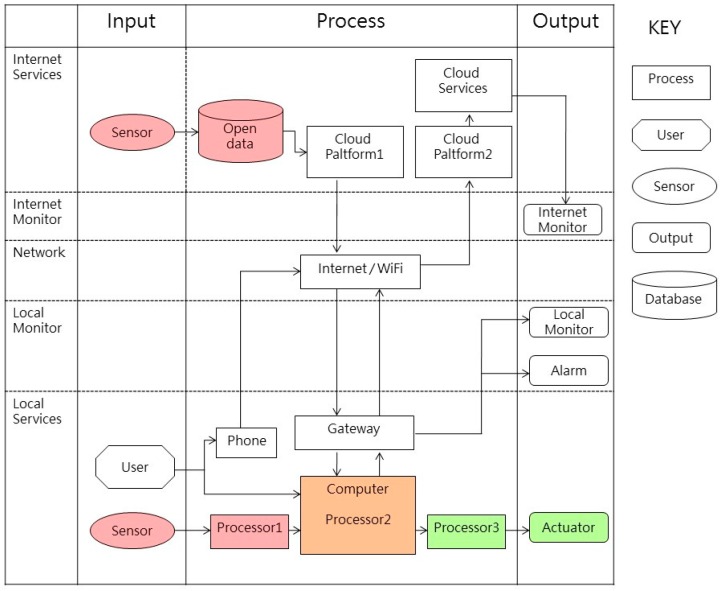
Block diagram of indoor air quality control system.

**Figure 9 sensors-19-01206-f009:**
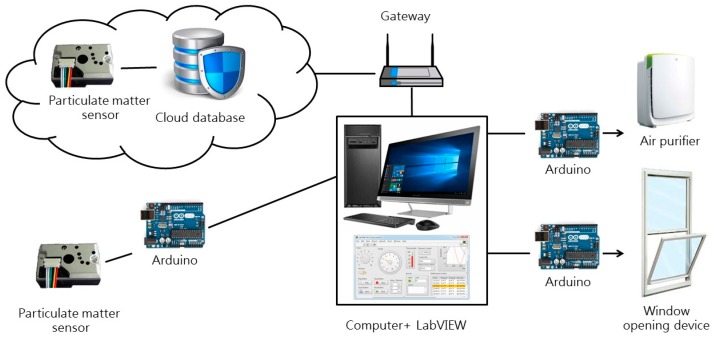
Schematic diagram of system connections.

**Figure 10 sensors-19-01206-f010:**
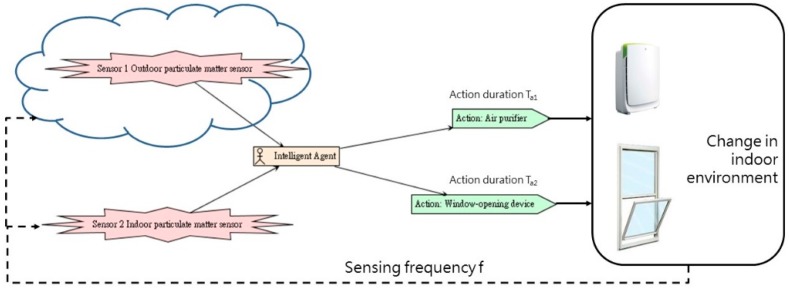
Individual intelligent agent module.

**Figure 11 sensors-19-01206-f011:**
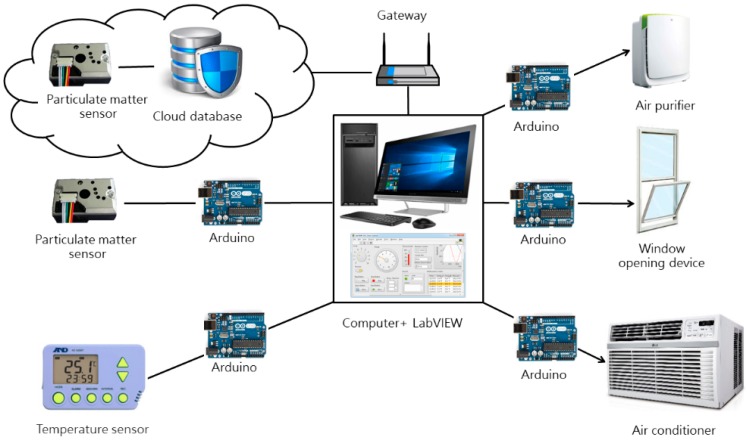
A multi-agent cooperation system.

**Figure 12 sensors-19-01206-f012:**
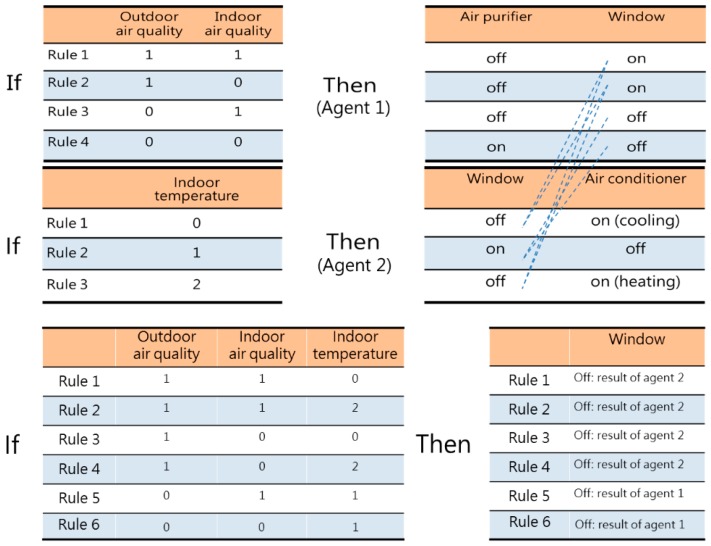
Rule table design, analysis of possibilities of conflict, and conflict resolution methods.

**Figure 13 sensors-19-01206-f013:**
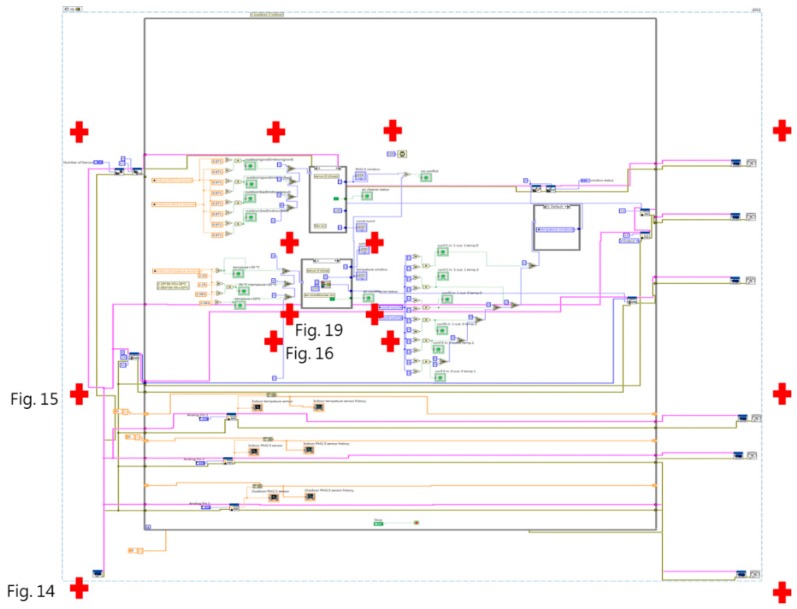
Full view of the LabVIEW multi-agent program.

**Figure 14 sensors-19-01206-f014:**
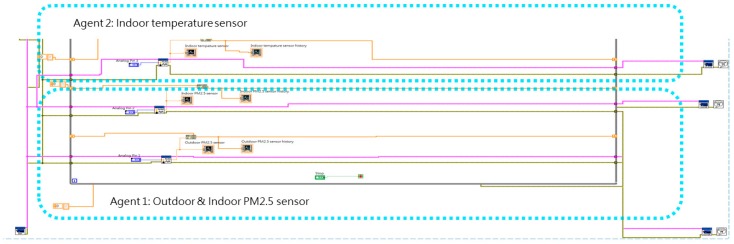
Sensor portion of multi-agent program.

**Figure 15 sensors-19-01206-f015:**
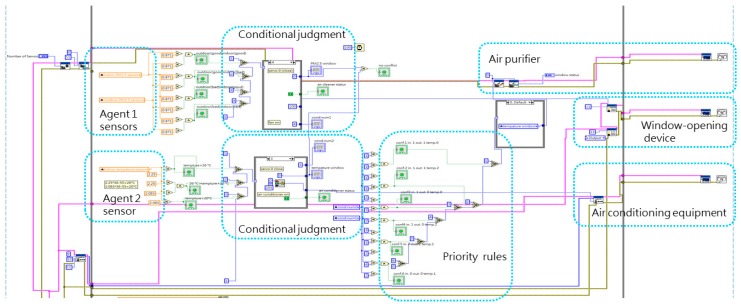
Intelligent agent and actuator portions of the LabVIEW program.

**Figure 16 sensors-19-01206-f016:**
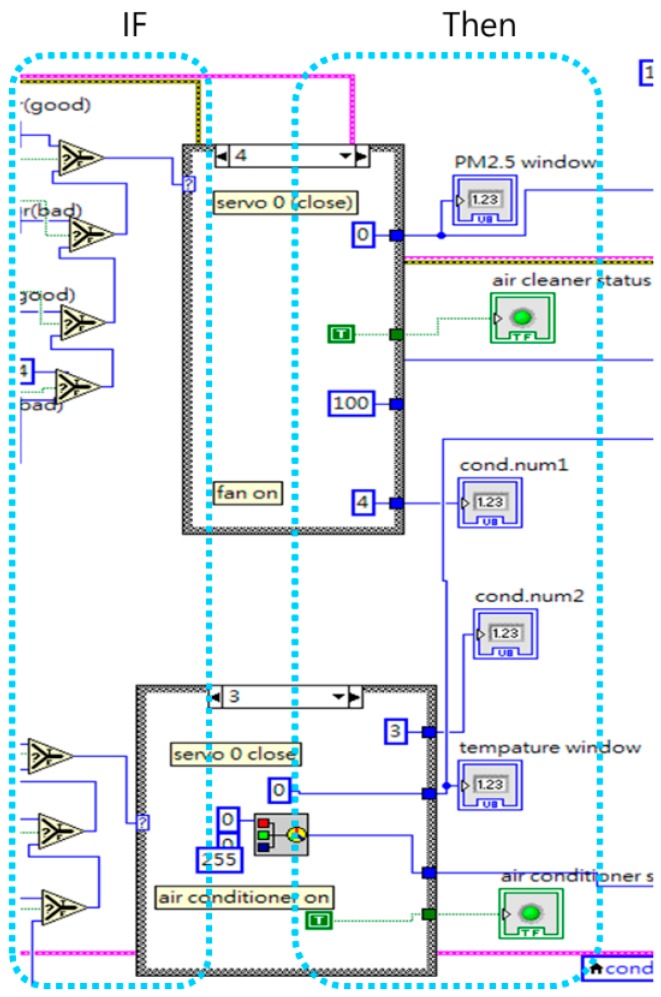
Rule-based case structure.

**Figure 17 sensors-19-01206-f017:**
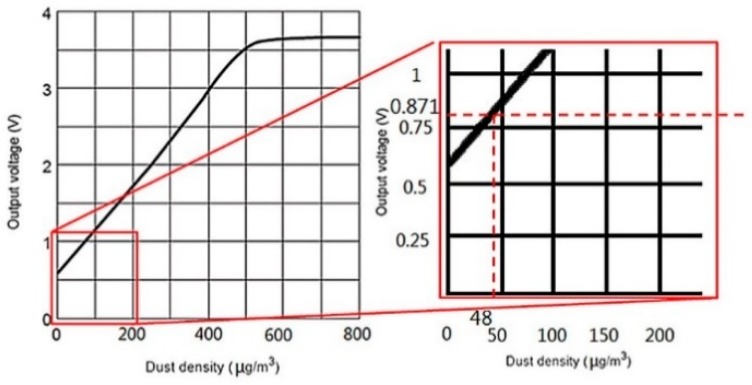
Diagram of particulate matter sensor data and voltage function [33].

**Figure 18 sensors-19-01206-f018:**
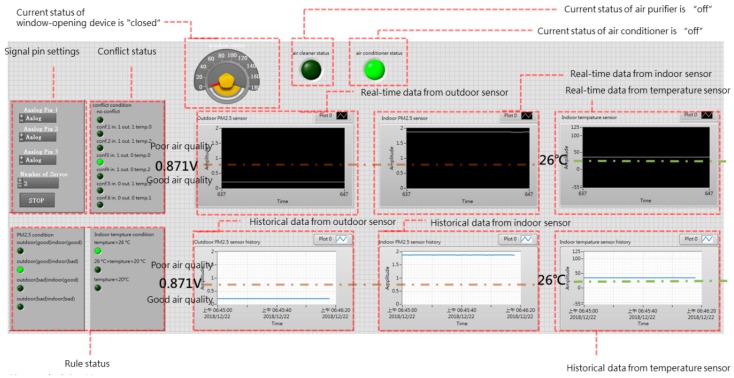
Explanation of user interface.

**Figure 19 sensors-19-01206-f019:**
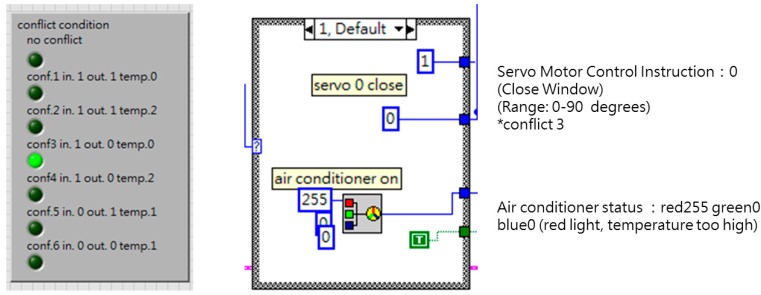
Rule 3 and its case structure.

**Figure 20 sensors-19-01206-f020:**
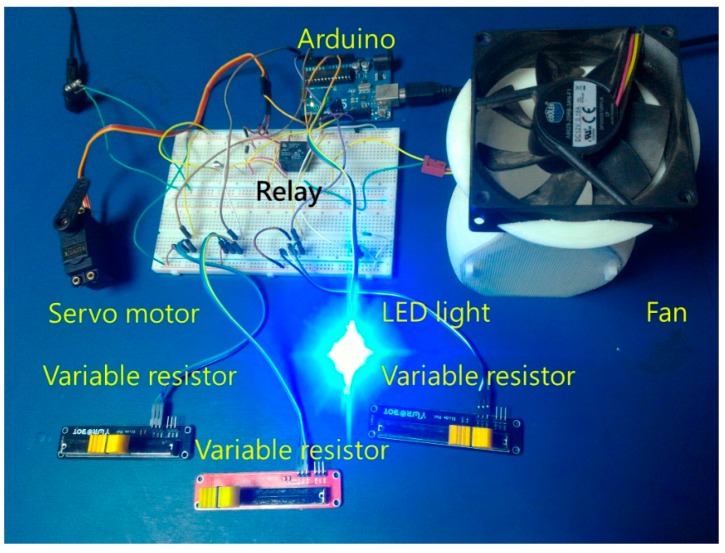
View of experimental devices.

**Table 1 sensors-19-01206-t001:** Fine particulate matter (PM_2.5_) indicator values and activity recommendations [30].

Indicator Grade	Classification	PM2.5 Concentration (μg/m^3^)	General Public Activity Recommendations	Activity Recommendations for Sensitive Groups
1	Low	0–11	Normal outdoors activities	Normal outdoor activities
2	Low	12–23		
3	Low	24–35		
4	Moderate	36–41	Normal outdoors activities 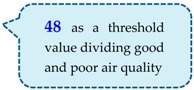	Adults and children with heart, respiratory, and cardiovascular disease should consider reducing physical exertion if they feel symptoms, and should reduce outdoor activities in particular.
5	Moderate	42–47		
6	Moderate	48–53		
7	High	54–58	Persons encountering discomfort, such as painful eyes, coughing, or throat pain, should consider reducing outdoors activity.	1. Adults and children with heart, respiratory, and cardiovascular disease should reduce physical exertion, and should reduce outdoor activities in particular. 2. Elderly persons should reduce physical exertion.
8	High	59–64		
9	High	65–70		
10	Extremely high	≧71	Persons encountering discomfort, such as painful eyes, coughing, or throat pain, should reduce physical exertion, and should reduce outdoor activities in particular.	1. Adults and children with heart, respiratory, and cardiovascular disease, and elderly persons, should avoid physical exertion, and should reduce outdoor activities in particular. 2. Persons with asthma may need to increase the frequency of inhalant use.

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
