# Peer review of "Use of Multi-Agent Theory to Resolve Complex Indoor Air Quality Control Problems"

_sensors, 2019, doi:10.3390/s19051206_

Round 1
Reviewer 1 Report
Authors propose in this paper agent system to make decisions in accordance with pre-written rules to construct and indoor air quality control system and conflict resolution mechanism, which will serve to maintain a healthy and comfortable indoor environment.
To achieve the purposed objective, Arduino open source microcontroller system to collect air quality data and turn on building equipment in order to improve indoor air quality with a graphic control program LabVIEW has been developed.
Authors have carried out a well-designed case study which aims to prove the initial hypothesis.
In general, it is a well structured article, about an interesting field and with good results.
[Minor]
-Maybe I would add a description of the results at the end of the abstract.
-The introduction section is wide but It might tend to introduce too basic concepts. It is advisable to complete this section with works that perform reviews of the state of the art, so as to support the decision to take multi-agent systems as the chosen methodology.
González-Briones, A., De La Prieta, F., Mohamad, M., Omatu, S., & Corchado, J. (2018). Multi-agent systems applications in energy optimization problems: A state-of-the-art review. Energies, 11(8), 1928.
Briones, A. G., Chamoso, P., & BARRIUSO, A. (2016). Review of the Main Security Problems with Multi-Agent Systems used in E-commerce Applications. ADCAIJ: Advances in Distributed Computing and Artificial Intelligence Journal, 5(3), 55-61.
-A cross-validation is required in order to validate the results with a brief detail about the sample, alpha, etc. You should have a look works like:
González-Briones, A., Prieto, J., De La Prieta, F., Herrera-Viedma, E., & Corchado, J. M. (2018). Energy optimization using a case-based reasoning strategy. Sensors, 18(3), 865.
Miki, U. E. N. O., Suenaga, T., & Isahara, H. (2017). Classification of Two Comic Books based on Convolutional Neural Networks. ADCAIJ: Advances in Distributed Computing and Artificial Intelligence Journal, 6(1), 5-12.
CASTRO, J., & MARTI-PUIG, P. (2014). Real-time Identification of Respiratory Movements through a Microphone. ADCAIJ: Advances in Distributed Computing and Artificial Intelligence Journal, 3(3), 64-75.
[Major]
-The references should be update and more recent. More references should be included to support the research carried out, both in the introduction and state of the art as in the part of the proposal.
-Figure 1 and 18 do not have a proper resolution and are difficult to read. It is recommended to change these figures to other figures with better resolution.
The work which have been carried out is significant, addresses interesting issues and is related to the SENSORS JOURNAL topics of interest.
The used English is correct.
Author Response
Comments and Suggestions for Authors
Authors propose in this paper agent system to make decisions in accordance with pre-written rules to construct and indoor air quality control system and conflict resolution mechanism, which will serve to maintain a healthy and comfortable indoor environment.
To achieve the purposed objective, Arduino open-source microcontroller system to collect air quality data and turn on building equipment in order to improve indoor air quality with a graphic control program LabVIEW has been developed.
Authors have carried out a well-designed case study which aims to prove the initial hypothesis.
In general, it is a well-structured article, about an interesting field and with good results.
Answer:
Thank you for your affirmation.
The author has published a series of journal articles on related topics (listed below), which solve some of the theoretical and practical problems in the past. However, this study uses Labview and Arduino technologies to integrate theory and practice and get feasible breakthroughs.
1. Chen, S. Y.; Chang S. F. A review of smart living space development in a cloud computing network environment. Comput.-Aided Des. Applic. 2009 6, 513–527. https://doi.org/10.3722/cadaps.2009.513-527
2. Chen, S. Y. The study of applying agent-based theory to adaptive architectural environments- Smart skin as an example, National Cheng Kung University Doctoral Dissertation, 28 July, 2007. http://dx.doi.org/10.6377%2fJA.200709.0095
3. Chen, S. Y.; Hu, Z. L.; A smart grid-based home energy-aware system, Sens Mater. 2017 29, 1513–1522. http://dx.doi.org/10.18494/SAM.2017.1660
4. Chen, S. Y., S. F. Chang, S. F.; Chang, Y. F. 2010/06, Exploring a designer-oriented computer aided design interface for smart home device. Comput.-Aided Des. Applic. 2010 7, 875–888. http://dx.doi.org/10.3722/cadaps.2010.875-888
5. Chen, S. Y.; Huang, J. T. Wen, S. L; Feng, M. W. Optimized control of indoor environmental health - the example of the Fu-An memorial building, Comput.-Aided Des. Applic. 2013 9, 733-745. https://www.tandfonline.com/doi/abs/10.3722/cadaps.2012.733-745
6. Chen, S. Y.; Huang, J. T. A Smart Green Building: An Environmental Health Control Design. Energies 2012 5, 1648–1663. https://doi.org/10.3390/en5051648
7. Chen, S. Y. Use of neural network supervised learning to enhance the light environment adaptation ability and validity of Green BIM, Comput.-Aided Des. Applic. 2018 15, 831-840. https://doi.org/10.1080/16864360.2018.1462566
[Minor]
-Maybe I would add a description of the results at the end of the abstract.
Answer:
Thank you for your comments. According to the conclusion, the end of the abstract is revised to:
The implementation verifies the feasibility of applying multi-agent theory to air quality control systems, and an Individual intelligent agent has the basic ability to resolve their own conflicts autonomously. However, when there are multiple factors and user status are simultaneously involved in the decision-making, it is difficult for the system to exhaust all conflict conditions, and when context control surpassing the restrictions of binary logic rule-based reasoning, it is necessary to change the algorithm and redesign the system.
-The introduction section is wide but It might tend to introduce too basic concepts. It is advisable to complete this section with works that perform reviews of the state of the art, so as to support the decision to take multi-agent systems as the chosen methodology.
Answer:
Thank you for your guidance.
This paper strengthens the literature review to prove that the application of multi-agent theory is not only an innovative concept but also a feasible technology. The start of the literature review is revised to:
In the ubiquitous computing era of the artificial intelligence (AI), the Internet of Things (IoT), and the big data technologies booming developments, how to build an agent-based smart living space which is capable of supporting sustainable energy conservation as well as maintaining health and comfortable environment for housing occupants has become an important research topic. [4-6] Autonomous intelligent agents and their society are capable of sensing, calculating, actuating, and communicating. The major dimensions of research include (1) Perception: Various environmental sensors, and signal processing. [7-9] (2) Computation: Algorithms include binary logic rule-based reasoning, expert system, artificial neural network learning algorithm, fuzzy neural network algorithm capable of reasoning and learning, and other optimization algorithms. [10,11] (3) Actuation: Agent behavior patterns include: An individual agent’s autonomous behaviors, including re-action, pro-action, and interactive. As well as the relationships of the agent society, including hierarchy, homogeneity, heterogeneity, and complementarity. [12] And actuators, including various building components and environmental control equipment. [13] (4) Communication: Transmission methods include wired network, wireless network, [8,9] micro-grids, [14] smart grids, [15] and communication protocol. (5) Integrated applications include: Programmable Controller, user interface (UI) design, [16,17] intelligent building design, [18,19] planning for smart city. And other derivative issues include: E-commerce applications, [20] information security, [20] optimization management, [21] post-use evaluation, [22] verification of system validity and reliability, [23] and more.
This paper proposes building multi-agent based smart living space with pro-active building components and environmental control equipment to improve the insufficient performance of a passive building and to achieve the maintenances of the purposes of sustainable energy conservation and occupants’ health and comfort simultaneously. Intelligent agent theory and indoor air quality indicators and guidelines are described as follows:
-Cross-Validation is required in order to validate the results with a brief detail about the sample, alpha, etc. You should have a look works like:
Answer:
Thank you for your guidance and help. The literature provided for demonstrating cross-validation by the reviewer has converged in the second section of the literature review. In my next phase of agent-based research, when multiple agents face the problem of "user preferences" in context control, binary logic rule-based reasoning is limited, and machine learning theory is needed at that time. Therefore, user behavior must be carefully recorded, including preferences for the machine control environment, and manually adjusting the values of the environmental control device afterward, and extracting this information as training data for machine learning. This article is limited to the length of the paper and considers the priority of dealing with those issues, therefore focuses on presenting the feasibility of multiple agents applied to environmental control with the binary logic rule-based reasoning using Arduino and LabView, and discusses the problems presented.
[Major]
-The references should be update and more recent. More references should be included to support the research carried out, both in the introduction and state of the art as in the part of the proposal.
Answer:
Thank you for your guidance.
This article strengthens the literature review
-Figure 1 and 18 do not have a proper resolution and are difficult to read. It is recommended to change these figures to other figures with better resolution.
The work which has been carried out is significant, addresses interesting issues and is related to the SENSORS JOURNAL topics of interest.
The used English is correct.
Answer:
Thank you for your reminder and affirmation
Due to layout restrictions, Figures 1 and 18 do have difficult to read. The author has tried to capture them with a higher resolution, and attach the original image files of 300dpi. Please, reviewers can understand and forgive.

Reviewer 2 Report
The authors develop a micro-controller based indoor air quality sensing system. Multi-agent theory is applied to analyze and decide the consequent actions. I have a few comments:
The number of reference papers is not sufficient. Please consider to add one new subsection in Section 2 (Literature Review) and review recent works in wireless sensor systems for indoor building environmental parameters, such as
[1] Qian Huang, Mao Chen, "occupancy estimation in smart building using hybrid CO2/light wireless sensor network," Journal of Applied Sciences and Arts, 2017.
[2] Francesco Salamone, Lorenzo Belussi, Ludovico Danza, Theodore Galanos, Matteo Ghellere, Italo Meroni, "Design and development of a nearable wireless system to control indoor air quality and indoor lighting quality," Sensors, vol. 17, no. 5, 2017.
[3] Qian Huang, Chen Mao, Yunfeng Chen, "A compact and versatile wireless sensor prottype for affordable intelligent sensing and monitoring in smart buildings," ASCE International Workshop on Computing in Civil Engineering, pp. 155-161, 2017.
[4] Francesco Ramos, Sergio Trilles, Andres Munoz, Joaquin Huerta, "Promiting pollution-free routes in smart cities using air quality sensor networks," Sensors, vol. 18, no. 8, 2018.
Before section 5, it is needed to provide a comparative study with other exisitng designs in the literature, to demonstrate the benefits and contributions of this work. Please add such a subsection in the revised version.
The authors should show that if no the proposed multi-agent system is utilized, how the perormance will be? Then, through comparing the cases with or without the proposed multi agent systems, the benefits of the proposed study is clear to readers.
Author Response
The number of reference papers is not sufficient. Please consider to add one new subsection in Section 2 (Literature Review) and review recent works in wireless sensor systems for indoor building environmental parameters.
[1] Qian Huang, Mao Chen, "occupancy estimation in smart building using hybrid CO2/light wireless sensor network," Journal of Applied Sciences and Arts, 2017.
[2] Francesco Salamone, Lorenzo Belussi, Ludovico Danza, Theodore Galanos, Matteo Ghellere, Italo Meroni, "Design and development of a nearable wireless system to control indoor air quality and indoor lighting quality," Sensors, vol. 17, no. 5, 2017.
[3] Qian Huang, Chen Mao, Yunfeng Chen, "A compact and versatile wireless sensor prototype for affordable intelligent sensing and monitoring in smart buildings," ASCE International Workshop on Computing in Civil Engineering, pp. 155-161, 2017.
[4] Francesco Ramos, Sergio Trilles, Andres Munoz, Joaquin Huerta, "Promiting pollution-free routes in smart cities using air quality sensor networks," Sensors, vol. 18, no. 8, 2018.
Before section 5, it is needed to provide a comparative study with other exisitng designs in the literature, to demonstrate the benefits and contributions of this work. Please add such a subsection in the revised version.
Answer:
Thank you for your guidance.
This paper strengthens the literature review to prove that the application of multi-agent theory is not only an innovative concept but also a feasible technology. And such a modification echoes the fifth section. The start of the literature review is revised to:
In the ubiquitous computing era of the artificial intelligence (AI), the Internet of Things (IoT), and the big data technologies booming developments, how to build an agent-based smart living space which is capable of supporting sustainable energy conservation as well as maintaining health and comfortable environment for housing occupants has become an important research topic. [4-6] Autonomous intelligent agents and their society are capable of sensing, calculating, actuating, and communicating. The major dimensions of research include (1) Perception: Various environmental sensors, and signal processing. [7-9] (2) Computation: Algorithms include binary logic rule-based reasoning, expert system, artificial neural network learning algorithm, fuzzy neural network algorithm capable of reasoning and learning, and other optimization algorithms. [10,11] (3) Actuation: Agent behavior patterns include: An individual agent’s autonomous behaviors, including re-action, pro-action, and interactive. As well as the relationships of the agent society, including hierarchy, homogeneity, heterogeneity, and complementarity. [12] And actuators, including various building components and environmental control equipment. [13] (4) Communication: Transmission methods include wired network, wireless network, [8,9] micro-grids, [14] smart grids, [15] and communication protocol. (5) Integrated applications include: Programmable Controller, user interface (UI) design, [16,17] intelligent building design, [18,19] planning for smart city. And other derivative issues include: E-commerce applications, [20] information security, [20] optimization management, [21] post-use evaluation, [22] verification of system validity and reliability, [23] and more.
The authors should show that if no the proposed multi-agent system is utilized, how the performance will be? Then, by comparing the cases with or without the proposed multi-agent systems, the benefits of the proposed study are clear to readers.
Answer:
Thank you for your review. The end of Section1 Motivation and goal is revised to:
From another perspective, because it is hard for people to pay attention to things continuously for long periods and people are not as sensitive as sensors which can aware the values harmful to the human body, therefore assistance from artificial intelligence is needed for the routine active control of indoor air quality. Intelligent agents and multi-agent societies are control systems relying on artificial intelligence to perform periodic sensing of the environment and take necessary corrective actions.

Round 2
Reviewer 1 Report
Paper is ok as it is.
Reviewer 2 Report
The quality of this paper has been improved. Yet, the resolution for figure 12-19 is low, so when printing out these figures, they look not very clear. Please replace them using high-resolution figures at least 300 dpi.